# Naltrexone Implant for Opioid Use Disorder

Amber N. Edinoff [1,*], Catherine A. Nix [1], Claudia V. Orellana [1], Samantha M. StPierre [2], Erin A. Crane [2], Blaine T. Bulloch [2], Elyse M. Cornett [3], Rachel L. Kozinn [4], Adam M. Kaye [5], Kevin S. Murnane [1,6,7] and Alan D. Kaye [3]

1   Department of Psychiatry and Behavioral Medicine, Louisiana State University Health Sciences Center Shreveport, Shreveport, LA 71103, USA; catherine.nix@lsuhs.edu (C.A.N.); claudia.orellana@lsuhs.edu (C.V.O.); kevin.murnane@lsuhs.edu (K.S.M.)
2   School of Medicine, Louisiana State University Health Sciences Center Shreveport, Shreveport, LA 71103, USA; SMP001@lsuhs.edu (S.M.S.); ean001@lsuhs.edu (E.A.C.); btb001@lsuhs.edu (B.T.B.)
3   Department of Anesthesiology, Louisiana State University Health Sciences Center Shreveport, Shreveport, LA 71103, USA; elyse.bradley@lsuhs.edu (E.M.C.); alan.kaye@lsuhs.edu (A.D.K.)
4   Department of Anesthesiology and Pain Management, University of Texas Southwestern, Dallas, TX 75390, USA; rachel.kozinn@phhs.org
5   Department of Pharmacy Practice, Thomas J. Long School of Pharmacy and Health Sciences, University of the Pacific, Stockton, CA 95211, USA; akaye@pacific.edu
6   Department of Pharmacology, Toxicology & Neuroscience, Louisiana State University Health Sciences Center Shreveport, Shreveport, LA 71103, USA
7   Louisiana Addiction Research Center, Shreveport, LA 71103, USA
*   Correspondence: amber.edinoff@lsuhs.edu; Tel.: +1-318-675-8969

**Abstract:** The continued rise in the availability of illicit opioids and opioid-related deaths in the United States has left physicians, researchers, and lawmakers desperate for solutions to this ongoing epidemic. The research into therapeutic options for the treatment of opioid use disorder (OUD) began with the introduction of methadone in the 1960s. The approval of oral naltrexone initially showed much promise, as the drug was observed to be highly potent in antagonizing the effects of opioids while producing no opioid agonist effects of its own and having a favorable side effect profile. Patients that routinely take their naltrexone reported fewer days of heroin use and had more negative drug tests than those without treatment. Poor outcomes in OUD patients treated with naltrexone have been directly tied to short treatment time. Studies have shown that naltrexone given orally vs. as an implant at the 6-month interval showed a higher non-compliance rate among those who used oral medications at the 6-month mark and a slower return to use rate. There were concerns that naltrexone could possibly worsen negative symptoms seen in opiate use disorder related to blockade of endogenous opioids that are important for pleasurable stimuli. Studies have shown that naltrexone demonstrated no increase in levels of anxiety, depression and anhedonia in participants and another study found that those treated with naltrexone had a significant reduction in mental health-related hospitalizations. The latter study also concluded that there was no increased risk for mental health-related incidents in patients taking naltrexone via a long-acting implant. Although not yet FDA approved in the United States, naltrexone implant has shown promising results in Europe and Australia and may provide a novel treatment option for opioid addiction.

**Keywords:** opiates; opioid use disorder; naltrexone; substance use disorders



## 1. Introduction

The continued rise in the availability of illicit opioids and opioid-related deaths in the United States has left physicians, researchers, and law makers focused on novel solutions for this ongoing epidemic. Despite the institution of new policies and restrictions over recent years on prescription opioids for the management of non-cancer-related chronic pain, data have demonstrated a continued rise in opioid-related overdose deaths in the

United States [1]. In 2016, an alarming 64,000 people in the United States died from drug fatalities, representing a 20% increase from 2015 [2]. Of those drug-related fatalities, over 42,000 were related to opioid drugs [2]. These trends have continued in recent years, with data released by the Centers for Disease Control and Prevention showing an increase of 4.6% in 2019 to 70,980 deaths, including 50,042 involving opioids [3]. In this regard, not only do numerous studies show the increase in opioid-related mortality over recent years, an increase in morbidity and health care financial burden has led our government to act in initiating several programs in an attempt to end this epidemic.

The origin of the opioid epidemic is thought to have occurred in the late 1990s with the American Pain Society's efforts to recognize pain as the fifth vital sign and the approval of a sustained formulation of oxycodone [4]. Alongside these events, the Joint Commission (TJC) published standards for pain management in 2000, mandating physicians to provide adequate pain control to all patients, resulting in the marked increase in prescription opioid medications [4]. Since then, the number of prescribed opioids has quadrupled, with opioids now being one of the most highly prescribed drug classes, even though no study to date has established long-term safety and efficacy of opioid drugs in reducing chronic pain intensity and improving function [4]. What has been proven is that the unprecedented abundance of opioid pain prescriptions within the United States has led to an exponential use of health care resources and finances, estimated to be responsible for between $53 and $72 billion in cumulative costs annually [5]. In addition to the rise in opioid deaths and economic costs, increases in the incidence of infectious diseases such as hepatitis C and HIV as well as a surge in admissions to the neonatal intensive care unit, with diagnosis of neonatal abstinence syndrome, have all been observed consequences paralleling the opioid epidemic [4].

These consequences led to the Centers for Disease Control and Prevention (CDC) adding opioid overdose prevention to the list of the top five public health challenges, and the US Government declaring the opioid epidemic a public health emergency in 2017 [2,6]. Since then, many initiatives have been made by the US Food and Drug Administration (FDA) as well as the TJC to create new guidelines for effective pain management with a strict emphasis on the public health ramifications of opioid prescriptions [2]. The implementation of prescription drug monitoring programs (PDMPs) has effectively shown to contribute to a reduction in opioid prescriptions, but these downward trends have been accompanied by an increase in heroin use and overdose [5]. This has led the FDA to explore novel options to combat this epidemic, through the use of abuse-deterrent opioid technologies and medication-assisted therapies [4]. The idea of abuse-deterrent technology is to create a chemical or physical formulation within the opioid prescriptions that discourage the misuse of the medications by injection and insufflation; however, many of these approved formulations are not available in generic form and are not widely available yet [4].

The research into therapeutic options for the treatment of opioid use disorder (OUD) has been long underway, beginning with the introduction of methadone in the 1960s [4]. The approval of more drugs indicated in the treatment for OUD including buprenorphine and naltrexone has been and will continue to be crucial in combating the opioid epidemic. Much evidence has shown that these medications all reduce dependency, overdose mortality, and risk of infectious disease transmission, as well as increase the likelihood that a person will remain in treatment [7]. Although highly efficacious, adherence to treatments can be a major challenge, and the study of better formulations and implant devices will play a major role in long-term treatment success moving forward. Daily medication treatment has been thought to lead to this decreased adherence and the thought of a treatment that does not need to be taken daily, and thereby serving as a daily reminder of the addiction, would be of great value.

## 2. Chronic Opiate Use

Most of the time, a patient's use of opioids begins with a written prescription by a health care provider that is intended to manage acute or chronic pain. Regardless of the pain source, exogenous opioids shut down endogenous opioid production, such as endorphins. In some cases, over time, patients begin to defy the prescription instructions and, if restricted, will begin to obtain opioids from someone other than their health care provider. To manage chronic pain and satisfy patients for adequate reimbursement, the prescription of opioids has increased dramatically in the last two decades. In addition, many prescribers do not recognize the signs of abuse and fail to effectively address the underlying cause of pain. These factors, among others, have led to an increased incidence of dependency, overdose, related deaths, and addiction to heroin [8]. This opioid epidemic presents a need for the identification of risk factors and prevention strategies so that health care providers can minimize potential adverse outcomes of those requiring opioids for pain relief.

### 2.1. Opioid Dependency Risk Factors

Many risk factors for opioid dependency, chronic use, and overdose have been identified in recent research. A prevalent risk factor is simply overprescribing. Patients who are prescribed less than 100 morphine milligram equivalents (MME) are less likely to overdose in comparison to patients who receive over 100 MME [8]. Patient demographics also contribute to the risk of opioid dependency. Multiple studies have noted that low socioeconomic status, those with medical assistance such as Medicaid, and those in rural areas have an increased risk of abuse [8,9]. Furthermore, elderly patients, especially those over the age of 80, are at a higher risk of abuse. Female sex is also a biological risk factor. Prescribers should inquire about the past history of substance abuse as this increases the risk of future opioid overdose [10]. Modifiable risk factors include nicotine addiction in any form, preoperative use of benzodiazepines, and underlying chronic pain [9].

### 2.2. Opioid Dependency Outcomes

A major public health concern in the United States is the growing number of deaths due to suicide and overdose. These patient outcomes are correlated to the number of opioids used to treat pain. In fact, in 2017, 40% of all suicide and overdose deaths were related to opioid use [11]. Furthermore, the abuse of opioids is correlated with the use of heroin, whose death rates have increased 20.6% from 2014 to 2015 [12]. The correlation between suicide, overdose, and opioid dependency can be contributed to the deterioration of physical and mental quality of life found in those who use opioids. Additionally, users are found to have higher rates of conditions such as cellulitis, necrotizing fasciitis, endocarditis, and transmissible diseases. The outcomes of opioid dependency are not restricted to affecting only the user but also impose a significant family and societal burden. Abuse can disrupt family finances, routine, interactions, and leisure. Furthermore, those with OUD utilize more health care resources and cost the system millions of dollars a year [13].

### 2.3. Opioid Dependency Prevention

The first step to preventing abuse is for physicians to consider nonpharmacologic, nonopioid treatments before prescribing opioids. Additionally, when prescribing an opioid, physicians should educate patients about the risks and goals of opioid therapy. Initial doses should be the lowest that is effective and patients should be re-evaluated every 1 to 4 weeks before re-prescribing [8]. If abuse does occur, there are treatments in place to prevent overdose and continued addiction. Naloxone, an opioid receptor antagonist, can effectively reduce respiratory depression in patients who overdose [8]. Wheeler and colleagues showed that providing naloxone to laypeople is effective at preventing overdose deaths in addition to being safe and cost-effective [14]. This should be considered as a method of preventing mortality it was found that many laypeople are willing to be trained

to administer naloxone [15]. Training in opioid overdose intervention is of vital importance since patients who have been treated for an overdose are at increased risk for subsequent opioid overdose [16]. Opioid users who wish to receive treatment to prevent further use have a few pharmaceutical options. Methadone and buprenorphine are full and partial agonists of the opioid receptors, respectively. Methadone's long half-life and decreased drug-like effects produce less euphoria and withdrawal symptoms. Buprenorphine also reduces euphoria but does so through only partially activating mu receptors and also decreases withdrawal symptoms [17].

### 3. Current Treatment of Opioid Use Disorder (OUD)

OUD treatment is aimed at reducing opioid withdrawal symptoms (OWS), preventing relapse, and reversal of overdose [18]. OWS includes hyperalgesia, tremor, anxiety, depression, intense cravings, irritability, nausea, vomiting, diarrhea, insomnia, lacrimation, sweating, and rhinorrhea among many others. These symptoms stem from adaptations that occur related to the routine use of opioid medications. The mesolimbic system has been associated with the intense cravings, depression, and irritability seen in OWS [18]. Nausea, vomiting, and diarrhea are associated with mu-opioid receptors in the GI tract. Many of the physically dependence symptoms seen in OWS haves been tied to the locus coeruleus (LC) and its projections [19]. Opioid binding to mu receptors on LC neurons causes a decreased firing rate which results in decreased respiration, muscle tone, blood pressure, and drowsiness. Continuous inhibition of LC neurons results in an adaptive response to increase baseline activity that offsets opioid effects and causes OWS in the absence of opioids [19]. Table 1 summarizes the systems and associated symptoms associated with either opioid use or withdrawal.

**Table 1.** Body systems involved in the associated symptoms of either opioid use or withdrawal.

| System Involved | Associated Symptoms | Withdrawal or Use |
|---|---|---|
| Mesolimbic System | Intense cravings Depression Irritability | Withdrawal |
| GI | Nausea Vomiting Diarrhea | Withdrawal |
| Locus Coeruleus | Decreased respiration Decreased muscle tone Decreased blood pressure Drowsiness | Use |

The initial step in treatment for both discontinuation and the reduction in opioid dosing is management of OWS. OWS is managed based on the severity of a patient's OUD, which is determined by guidelines established in the *Diagnostic and Statistical Manual of Mental Disorders, Fifth Edition (DSM-5)* [18]. Management of OWS is performed medically. Opioid agonist therapy (OAT) is the preferred treatment for patients with moderate or severe OUD to alleviate OWS [20]. Methadone and buprenorphine are used for OAT. They are ideally started at the beginning of OWS and either kept at a low steady state or slowly tapered down over time. Methadone is a synthetic mu-opioid receptor agonist with a half-life of 28 h allowing for daily dosing [21]. Methadone reverses OWS by activating mu-opioid receptors, it decreases cravings for opioids and binds more avidly to the mu-receptors than used opioids which decreases their euphoric effects. Unlike abused opioids, methadone does not develop tolerance commonly and can be kept at a steady state for long-term management of OUD [22]. Buprenorphine is a semi-synthetic mu-opioid partial agonist and has weak partial agonist effects at the delta and kappa opioid receptors. If a patient takes buprenorphine shortly after using an opioid it will induce OWS. For this reason, patients should not use any opioid 12–24 h before taking buprenorphine [21].

Naltrexone is another medication that is used to prevent relapse in patients with OUD. Naltrexone is a mu-opioid receptor antagonist. This medication works by blocking the euphoria of opioids by binding and blocking the mu-receptor with a higher affinity [21]. Some patients with OUD might have contraindications to the use of OAT. Examples of these contraindications are mild OUD, patients who plan to be treated with naltrexone, and patients who desire opioid-free therapy [21]. For patients at the minimal dosage of OAT or who are contraindicated to start OAT, several nonopioid medications can be used to alleviate their symptoms. Lofexidine is an α-2 adrenergic agonist that is FDA approved for the treatment of OWS. α-2 adrenergic agonists alleviate many symptoms of OWS by acting as autoreceptor feedback inhibition on the LC pathways that cause noradrenergic hyperactivity [23]. Clonidine is another α-2 adrenergic agonist that has shown effective management of OWS that is comparable to lofexidine [24]. A number of techniques have been described and utilized successfully over the past two decades that employ clonidine, propofol, intubation, naloxone, and post-operative oral naltrexone to ultra-rapidly precipitate withdrawal under general anesthesia [25–29].

Other medications that are not FDA approved for the management of OWS have been used to alleviate more specific symptoms of OWS. Some examples include over-the-counter medications such as nonsteroidal anti-inflammatory drugs (NSAIDs) and acetaminophen, which can be used for hyperalgesia and musculoskeletal pain. Ondansetron, prochlorperazine, and metoclopramide are medications that can be used for nausea and vomiting. Bismuth can be used for diarrhea associated with OWS. Eszopiclone, zolpidem, or low doses of trazodone and doxepin can be used for insomnia [18]. Loperamide and Benzodiazepines have been used in the past for the management of diarrhea and anxiety, respectively; however, these medications have a risk for abuse in patients with OUD and should be used with caution [30].

Patients who present with opioid overdose are in need of emergency treatment due to life-threatening respiratory depression. Naloxone is the drug of choice for emergent reversal of opioid effects and is a mu-opioid receptor antagonist. It displaces opioids off the mu receptor for rapid reversal of opioid overdose. Naloxone also causes rapid OWS due to displacement from the mu receptor. Patients should be treated with nonopioid medications such as another α-2 adrenergic agonist to alleviate these symptoms [18].

## 4. Naltrexone Implant

Since FDA approval in 1984, the use of naltrexone in tablet form has helped shape the treatment of OUD [31]. The approval of oral naltrexone initially showed much promise, as the drug was observed to be highly potent in antagonizing the effects of opioids while producing no opioid agonist effects of its own and having a favorable side effect profile [32]. However, like many of the other oral formulations available for the treatment of OUDs, treatment success is strongly correlated with patient compliance, making daily oral formulations a hit or miss strategy. This led to the research and creation of sustained-release forms of naltrexone to increase compliance and ultimately improve the overall effectiveness of the treatment. In 2010, the FDA approved a monthly depot injection of naltrexone for relapse prevention of both OUD and alcohol use disorder (AUD) [4]. Despite the fact this sustained-release depot formulation decreases the need for patients to take a daily oral medication, its efficacy still relies on patients returning once a month for repeat injections, making treatment with even longer-acting formulations a more desirable alternative [33].

Although not yet FDA approved in the United States, the efficacy and safety of naltrexone implants are being studied in Australia and various parts of Europe [34]. Many of the implants being studied have varying degrees of duration but could be efficacious for up to 6 months, dramatically reducing the need for continued daily or monthly compliance [33]. A naltrexone implant manufactured in Australia was implanted subcutaneously into the abdomen under local anesthetic and was designed to release a total of 2.3 g of naltrexone [33]. Researchers found that this dosing was able to maintain naltrexone levels at or above 2 ng/mL for approximately 5.5 months when standardized to a 70 kg person [33,35].

Further research is needed to examine the safety and efficacy of these implants, but current studies show that it may be a promising option in the treatment of OUDs moving forward.

*Mechanism of Action*

Naltrexone is a competitive inverse agonist of opioid receptors in the central nervous system (CNS), with the highest affinity for the mu-opioid receptor [36]. As an inverse agonist, Naltrexone binds to the same receptor-binding site as opioid agonists and antagonizes the effects of that agonist as well as exerts the opposite effect by suppressing the receptor signaling that would occur with the binding of an agonist [37]. Naltrexone may also reverse, although not fully, the effects induced by partial agonists. It is a cyclopropyl derivative of oxymorphone with a structure similar to naloxone and nalorphine [38]. Naltrexone has few intrinsic actions other than its blockage of mu-opioid receptors; however, it has been shown to produce pupillary constriction by an unknown mechanism [39]. By competitively occupying opioid receptors within the central nervous system, naltrexone may block the effects of endogenous opioid peptides [40]. The mechanism in which naltrexone is useful in the treatment of AUD is not fully understood, but it is believed to have to do with the blockage of endogenous opioids [36]. Naltrexone may also modify the hypothalamic–pituitary–adrenal axis, aiding in the suppression of alcohol consumption [41]. The competitive inhibition of the mu-opioid receptors leads to antagonization of many of the subjective and objective effects that opioids produce including respiratory depression, euphoria, drug craving, and miosis [42]. Inhibition of opioid receptors by naltrexone can potentially be surmounted by the administration of opioids and may lead to non-opioid receptor-mediated effects such as histamine release [36]. The use of naltrexone is not associated with the development of dependence or tolerance, and it has few side effects even when taken for extended periods [32]. Naltrexone has an overall favorable side effect profile; however, it does have a black box warning for causing hepatotoxicity when given in excess doses and therefore is contraindicated in patients with acute hepatitis or liver failure [43]. In patients that are physically dependent on opioids, naltrexone use will precipitate withdrawal symptoms, and therefore it should not be used before the completion of a medically supervised withdrawal from opioids [44].

## 5. Pharmacokinetics/Pharmacodynamics

Naltrexone is an opioid receptor antagonist used for the management of addiction-based cravings of opioids and heroin. It is available in two sustained-release forms: intermuscular injection and surgically implantable pellets. Injections are administered in the gluteus muscle and produce detectable levels of naltrexone for 4 weeks. The implantable pellets can produce variable detectable levels of naltrexone depending on the type that is used. An implant that was developed in Russia can sustain therapeutic naltrexone levels for 2 to 3 months [45]. A single Australian implant provides sufficient therapeutic naltrexone levels for 2 to 4 months while a double Australian implant extends implant efficacy to 5 to 6.5 months [46]. With naltrexone pellets, serum levels are highest within a day or two of implantation [47,48]. Over time, the concentration of naltrexone trends downward, with precise concentrations varying between individuals receiving identical implants at similar time points [47]. In one mouse study, pellets released 40% of their naltrexone in 24 h with the remainder naltrexone released over the next 168 h. Despite reaching maximum plasma levels of naltrexone in 24 h, the remaining time-released naltrexone sustained therapeutic plasma levels [48]. Similar trends were observed in human studies, with an initial spike and subsequent gradual decline in plasma naltrexone. The time-release period of naltrexone from its implant seems to be dose-dependent, with single implants declining to 1 to 2 ng/mL after 1 to 3 months compared to 3 to 5 months with a double implant [46]. Based on these data, new formulations of implantable pellets with stable a more time-release and longer periods of efficacy beyond 6 months would benefit the opioid-dependent population [49].

## 6. Safety and Efficacy

### 6.1. Safety

Naltrexone has a well-established safety profile [21]. It has some minor side effects such as headache, nausea, vomiting, and dysphoria, and these symptoms are typically not severe enough to stop the medication [50]. There is one black box warning for naltrexone which is hepatotoxicity [51]. The prevalence of the side effect among patients has a wide variance in different studies [51,52]. The most common lab abnormality seen in AUD is elevated liver enzymes due to the toxic effects of ethanol on the liver [51]. Patients with a history of alcohol abuse or other causes of liver damage need to be assessed before starting naltrexone [51]. Patients with mild to moderate cirrhosis or chronic hepatitis can take naltrexone safely with routine monitoring [53]. Patients with severe cirrhosis, acute liver injury, or acute hepatitis are contraindicated from starting the medication [51]. In patients that are physically dependent on opioids, naltrexone use will precipitate withdrawal symptoms, and therefore it should not be used prior to the completion of a medically supervised withdrawal from opioids [44]. For this reason, naltrexone is contraindicated in patients taking buprenorphine, methadone, or any other opioid for medical purposes because it will negate the effects of the opioid agonists [54]. It may precipitate opioid withdrawal symptoms in a patient who has recently used an opioid agonist [21]. While OWS is not lethal on their own, acute exacerbations of OWS can be life-threatening due to volume depletion from vomiting and diarrhea [18]. For this reason, it is important to abstain from opioid use before beginning naltrexone therapy. Studies have shown that if a patient has used heroin or any other opioid within the last 7–10 days, they should not be started on naltrexone therapy [54]. If a patient is given naltrexone too soon and severe OWS are precipitated, they can be managed with buprenorphine, $\alpha$-2 adrenergic agonists, or other medications targeted at specific symptoms [55].

### 6.2. Efficacy

Naltrexone has been proven to effectively block opioid receptors from being stimulated and prevent patients with OUD from experiencing the typical symptoms of using an opioid [21]. This makes it very useful in managing patients with OUD by completely blocking the effects of heroin or other abused opioids making relapse to other opiates very unlikely. The benefit of the medication can be considered two-fold causing positive reinforcement against the use of opioids due to negative reinforcement from monetary loss with no gain [56]. Patients that routinely take their naltrexone reported fewer days of heroin use and had more negative drug tests than those without treatment [57]. The challenge with oral naltrexone therapy, as stated earlier in this manuscript, is that it requires daily compliance. Many patients that struggle with OUD will abstain from their naltrexone use to use opioids again [33]. Naltrexone does not produce any adverse effects when it is stopped and does not relieve OWS when it is initiated so it is very easy for a patient with OUD to stop using the medication [58]. Unlike naltrexone, methadone and buprenorphine do have adverse events when stopped and relieve OWS when taken, therefore compliance is improved [58]. Treatment adherence has been historically poor for oral naltrexone, some studies have shown a retention rate lower than 20% at 6 months. A meta-analysis performed in 2011 showed that oral naltrexone therapy was no better than placebo treatment [59]. This is particularly concerning because opioid tolerance is reduced over time while a patient is on naltrexone, if a patient stops taking their naltrexone and abuses heroin or another opioid they are at an increased risk for overdose, respiratory depression, and death [60]. As with most medications used in treating addiction, naltrexone studies show that patient outcomes improve with social support, after-care counseling, compliance strategy training, and psychotherapy [58]. This shows that while naltrexone can be effective in the treatment process for OUD it should not be used alone without proper social structuring for the patient and any measures that can be taken to increase compliance will benefit the patient.

Poor outcomes in OUD patients treated with naltrexone have been directly tied to short treatment time, studies have proven that when patients are in treatment for long periods

with naltrexone, they have variable outcomes [58]. A solution to the low compliance of oral naltrexone is the use of a sustained-release solution that is either injected or surgically implanted. Sustained-release preparations contain either compressed naltrexone or a naltrexone/polymer/copolymer combination [61]. These solutions are typically injected subcutaneously and maintain therapeutic blood levels of naltrexone for 4 to 6 weeks depending on dosing [61].

A study in 2004 that measured the blood levels of naltrexone after the use of the 1.7 g and 3.4 g naltrexone injectable has shown that the injectable can maintain above therapeutic levels for 3 and 6 months, respectively [60]. This is beneficial for the patient because they do not have to be taken daily oral medications, which eases their path to recovery [58]. One study measuring outcomes of naltrexone given orally vs. as an implant at the 6 month interval showed a higher non-compliance rate among those who used oral medications at the 6 month mark [33]. It also showed that patients who used oral naltrexone returned to heroin use sooner than those that had used the implant (median [SE],115 [12.0] days vs. 158 [9.4] days) [33]. This shows a clear benefit of injectable naltrexone over oral. There seems to be a direct relationship between compliance and the length of time that the naltrexone injectable maintains therapeutic levels [33].

Another study performed in 2005 that measured outcomes at 12 months showed a statistically significant improvement in patient outcomes in patients using implants vs. taking oral medications [52]. Interestingly, it also showed proof that the longer a patient is on the naltrexone medication that their long-term outcomes will be better. For example, patients in the oral group who did not have a relapse had taken naltrexone for an average of 4 months during the study. Those who did have a relapse in the oral group had stopped taking their medicine on average within the first two weeks of the study [52]. The study also showed that patients' understanding of their medication affects outcomes. Of those in the subcutaneous injection group, the patients who did not relapse within the 12 month window believed the implant lasted longer, upwards of 6 months than those who did relapse which believed the implant only was effective for 3 to four months [52]. The 12 month study also showed data on the effects of oral vs. implanted naltrexone on self-confidence at the 6 and 12 month mark. It showed that before treatment there was no statistical difference in patient's self-confidence levels, but there was a statistically significant increase in self-confidence in patients in the oral group at 6 months. Interestingly, there was no significant difference at the 12 month mark between the two. The study indicated that the improved self-esteem was maintained in the oral group, but the implant group rose to a similar level by the 12 month interval [52].

A study was performed in Australia measured the efficacy of oral naltrexone vs. an implant, but it focused more on blood levels of the medication and cravings for heroin or other abused opioids [61]. The study showed that blood naltrexone concentrations below 0.5 ng/mL were associated with a much higher risk of increased use of heroin and sensation of withdrawals. The study showed that at a concentration of 1 ng/mL of naltrexone patients had a 35% reduction in the odds of them using heroin. Patients with a blood naltrexone concentration of 3 ng/mL were associated with a very low risk of relapse; however, at dosages higher than 3 ng/mL there was no statistical evidence of increased effectiveness of naltrexone's ability to reduce cravings and prevent relapse [61]. The study showed that the patients that received the implanted naltrexone had a lower number of cravings for heroin and with it a lower rate of relapse. The implant group had one-fifth the risk of using heroin when compared to the oral group in this study [61]. One variance between the two groups that was interesting was that there were statistically significant lower cravings in the implant group with larger dosages of naltrexone, up to 3 ng/mL [61]. The oral naltrexone group had different results which showed no relationship between increased blood levels of naltrexone changing the amount cravings for heroin. The study concluded that an appropriate blood level for naltrexone for the treatment of OUD is 1 to 3 ng/mL, any treatment bellow 0.5 ng/mL is insufficient for treatment, and that implanted naltrexone might be more efficacious than oral naltrexone since it reduces non-compliance,

as well as keeping a more consistent blood level of the medication for an extended period of time which appears to have effects on patient's cravings [61]. Another assessment made by the study is that there is a strong association between the intensity of cravings a patient has and imminent relapse. The study suggested screening for the severity of cravings could be used to determine if a patient needs more acute management of their OUD to prevent relapse in the near future.

Another aspect of naltrexone's efficacy and safety that has been studied is its effect on mental health. This is particularly important in treatment for patients with OUD because they have an increased rate of depression, suicidal ideation, and anxiety [62]. There were concerns that naltrexone could worsen negative symptoms seen in OUD due to the blockade of endogenous opioids that are important for pleasurable stimuli [62]. One study followed a group of patients with OUD and measured their anxiety, depression, and anhedonia before and after treatment with naltrexone. The study showed that patients with OUD had higher baseline levels of anxiety, depression, and anhedonia when compared to the general public, but it decreased back down to normal levels 1–2 months into treatment with naltrexone. This showed the opposite of what some had predicted would be a problem with naltrexone therapy.

Another study evaluated the number of hospital events patients had pre- and post-treatment with naltrexone implants to see if there is any correlation between the implant and mental health outcomes [63]. The research showed that there was not an increased rate of mental health-related hospitalizations in patients who were treated with naltrexone [35]. There was a significant reduction in the number of hospitalizations for non-substance mental disorders, substance-related mental disorders, and all-cause mental disorders in patients when compared pre- and post-treatment groups. Further stratification showed that this decrease in mental disorder-related hospitalizations was most prevalent in the male population of patients studied and was often nonsignificant in the female population [35]. There was a category of mental disorder that had no significant difference between pre- and post-treatment, regardless of sex, which was mood-related disorders. The study concluded that there was no increased risk for mental health-related incidents in patients taking naltrexone via a long-acting implant. Something the study did bring to light was in all the subgroups studied, young females with a history of mental illness had the highest rate of future mental illness-related events and would be considered a "high risk" group when treating OUD with naltrexone. Another observation made by this study was an increased risk for patients to have mental health problems in the future based on how long a patient has been using heroin and how much they had been using [63].

While there is a statistically significant improvement in compliance, it is still problematic with the injectable form. Reducing the number of times a patient has to take their medication will help increase their chances of achieving long-term sobriety goals, but only to a certain degree. A 2006 study showed that 30–40% of patients on the long-acting injectable of naltrexone failed to return for follow-up dosing [64]. Table 2 summarizes the safety and efficacy of naltrexone implant.

**Table 2.** Clinical Safety and Efficacy.

| Author (Year) | Groups Studied and Interventions | Results and Findings | Conclusions |
|---|---|---|---|
| Hulse G. et al. (2009) [33] | Phase IV Randomized, double-blinded, double placebo-controlled trial with a 6 month follow up period. Participants either received naltrexone implant with placebo oral medicine or a placebo implant with oral naltrexone. Main outcomes studied were blood naltrexone levels, return to heroin or other abused opioid use, opiate overdoses and other adverse events. | More participants in the oral vs. implant group had naltrexone levels below 2 ng/mL in their blood in the first ($p = 0.001$) and second ($p = 0.01$) months. More people in the oral group returned to heroin use at the 6 month mark ($p = 0.003$) and they had done so earlier than the implant group (median (SE), 115 (12.0) days vs. 158 (9.4) days). | Naltrexone implant reduces relapse to regular heroin use better than oral naltrexone and was not associated with major adverse events. |

**Table 2.** *Cont.*

| Author (Year) | Groups Studied and Interventions | Results and Findings | Conclusions |
|---|---|---|---|
| Comer S. (2006) [64] | Phase IV randomized, double-blind, placebo-controlled, 8 week trial conducted at 2 medical centers. Participants were separated into placebo, or 192 mg or 384 mg dosage of depot naltrexone. All participants received relapse prevention therapy two times a week. Main outcomes measured were retention in treatment and percentage of opioid negative urine samples. | Retention in treatment was dose related with 39% in the placebo, 60% in the 192 mg, and 68% in the 382 mg group remaining at the end of the two-month mark. The mean number of days to dropout was lowest in the placebo group (27 days; 3.8 weeks), followed by the 192 mg of naltrexone group (36 days; 5.1 weeks), and the 384 mg of naltrexone group (48 days; 6.8 weeks). There was a significant difference in days to drop out between the placebo group and the 384 mg group ($p < 0.001$) and significant difference between the 384 mg group and the 192 mg groups ($p = 0.046$). | Long-lasting antagonist of the mu receptor is a feasible, efficacious, and tolerable treatment for OUD. |
| Hulse G. et al. (2004) [60] | Phase IV Retrospective clinical record review of blood sample analysis results. Main outcomes measured were plasma levels of medication, retention rate, and time to drop out. | The mean (+SE) length of follow-up was 197.1 (+30.5) days (minimum length of follow-up was 70 days and the maximum 333 days). In patients who used the 1.7 g implant the blood levels of naltrexone were above 2 ng/mL for 90 days and remained above 1 ng/mL for 136 days on average. In the 3.4 g implant Blood naltrexone levels remained above 2 ng/mL for 188 days and remained about 1 ng/mL for 297 days on average. | Conclusions by this record review were that blood levels of naltrexone were markedly higher in this study than what has been reported in other studies. They also conclude that these implants offer significant improvements over earlier naltrexone implants previously reviewed. |
| Colquhoun R. (2005) [55] | Phase IV cohort study following up patients that went through oral vs. implant naltrexone therapy for OUD. | The study showed a significant correlation with implant relapse and how long the participant though the implant was effective for ($p < 0.05$). There was also significant decrease in opiate relapse in the implant group when compared to the oral group ($p < 0.05$). | This study demonstrates the potential for implanted naltrexone to improve compliance rates, abstinence rates, and time in treatment when compared with oral naltrexone therapy. |
| Hulse G. et al. (2010) [61] | Phase IV randomized, double-blinded, double dummy design where participants either received an active implant and placebo oral medication or active oral medication and placebo implant. Major outcomes measured were level of cravings, number of heroin uses per week, and any illicit opioid use. | The implant naltrexone participants were less likely to use any opioids when compared with the oral group. There was a significant correlation between naltrexone blood concentration and decreased cravings in the implant group ($p = 0.0208$). There was not a significant decrease in cravings with increased blood naltrexone levels in the oral group. There was less cravings seen in the implant group vs. the oral group but the difference between these two groups was not significant. | Naltrexone implants were associated with reduced heroin cravings and relapse when compared to oral naltrexone. Effective treatment range for the medication is 1–3 ng/mL in the blood. |
| Ngo H. et al. (2006) [63] | Phase IV cohort study on heroin users treated with implanted naltrexone. Major outcomes measured were hospitalizations due to mental illness. | Patient's risk for hospitalization due to mental illness was not affected by naltrexone use. In fact, there was a statistically significant ($p < 0.05$) decrease in hospitalization rates for mental illnesses after naltrexone therapy except for in mood disorders where there was no significant difference pre- and post-treatment. | Naltrexone implant was not associated with an increased risk for mental illness-related hospitalizations. In most cases it was associated with a decrease in hospitalizations due to mental illness excluding mood disorders. |

## 7. Conclusions

With the continued rise in opioid-related deaths in the United States, health officials are desperate to come up with new treatment modalities and action plans to combat this crisis. The opioid epidemic began as a multi-faceted, complex issue and defeating the crisis will require a collaborative effort from communities across the country. Doctors, scientists, pharmacists, and law enforcement agencies are just a few of the many groups researching better ways to target this crisis and to ensure safe opioid prescribing with an overall decrease in the morbidity and mortality associated with opioid addiction. Some of the roles that the physician plays in combating the epidemic is to identify patients with risk factors of addiction prior to drug administration, educate patients on the dangers of opioid prescriptions, and help provide a safe route of detoxification for those patients

wanting to stop using opioids. Drugs such as naltrexone implant show promise in creating a better, safer way for patients who suffer from opioid addiction to maintain their sobriety. Naltrexone, a mu-opioid receptor antagonist, was first approved as an oral formulation for the treatment of AUD and OUD, but the need for the medication to be taken daily made it an unfavorable formulation and lead to the approval of the naltrexone monthly depot injection [4]. Although this medication has led to improved compliance and therefore improved efficacy of naltrexone, researchers have been working to create even better, longer-lasting formulations that will improve current treatment modalities for OUD. Although not yet FDA approved in the United States, naltrexone implant has shown promising results in Europe and Australia in the next wave of treatment options for opioid addiction. Research in Australia has shown that naltrexone implant is superior to oral naltrexone in reducing regular heroin use without causing significant events [33]. A phase IV cohort study performed in 2005 has found similar data and has shown that naltrexone implant has the potential to improve compliance rates, abstinence rates, and time in treatment when compared with oral naltrexone therapy [58]. These findings make naltrexone implants a promising drug in increasing long-term sobriety and decreasing the morbidity, mortality, and overall health care economic burden of OUD.

**Author Contributions:** A.N.E. and A.M.K. were involved in the conceptualization of the manuscript. A.N.E., S.M.S., E.A.C. and B.T.B. were involved in the writing of the manuscript. A.N.E., C.A.N., C.V.O., K.S.M., R.L.K., E.M.C., A.D.K. and A.M.K. were involved in manuscript editing. All authors have read and agreed to the published version of the manuscript.

**Funding:** This research received no external funding.

**Institutional Review Board Statement:** Not applicable.

**Informed Consent Statement:** Not applicable.

**Data Availability Statement:** Data supporting the results above can be found on pubmed.

**Conflicts of Interest:** None of the authors have any conflict of interest to report in this project.

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
