# Peer review of "Naltrexone Implant for Opioid Use Disorder"

_2035-8377, doi:10.3390/neurolint14010004_

Round 1

Reviewer 1 Report

Dear Authors,

Many thanks for submitting this paper.

It is of interest as it discusses a potential medication that could be employed in the treatment of OUD.

The article is good, and there are some paragraphs that are very helpful in advice e.g. 2.1 Risk factors, 2.2 Outcomes and 2.3 prevention, and as such I only have a few comments which would improve the article.

Page 2&3 there is lots of mentions of "opioid abuse", I think this is stigmatising and the use of "opioid dependence" or "opioid dependency" is preferable. Sorry there are multiple mentions so I have not given line numbers.

Line 211 - I think it should be the "emergency treatment" and not the "emergent treatment"

Table 2 - can you add the correct reference numbers to the articles mentioned in the table to make these easy to identify and relate to especially as there are a number from the same lead author.

I think with these adjustments the article is suitable for publication.

Many thanks

Author Response

Page 2&3 there is lots of mentions of "opioid abuse", I think this is stigmatising and the use of "opioid dependence" or "opioid dependency" is preferable. Sorry there are multiple mentions so I have not given line numbers.

            Answer: Thank you for pointing that out. We tried to edit this out during the final rounds of editing but missed some. We have correct that in these revisions.

Line 211 - I think it should be the "emergency treatment" and not the "emergent treatment"

            Answer: This has been corrected.

Table 2 - can you add the correct reference numbers to the articles mentioned in the table to make these easy to identify and relate to especially as there are a number from the same lead author.

            Answer: Absolutely! This is a great point and those reference numbers have been added to this revision.

Reviewer 2 Report

Firstly, I would like to congratulate the Authors for presenting such an interesting paper. It is well written and it tries to face the problem of opioid abuse.

However, before the publication, some issues need to be corrected.

  1. Naltrexone is an inverse agonist at opioid receptors, and this should be corrected and somehow detailed defined.
  2. Also, it should be noted that naltrexone may reverse, however not fully, the effects induced by partial agonists.
  3. In the section no. 5 the Authors provided some information regarding pharmacokinetics and/or pharmacodynamics profile of naltrexone, while presenting the examples of naltrexone implants, depending on the country of origin. It would be great if the Authors could demonstrate similarities and differences of the above mentioned in the table, including additional substances used by the producers.

Author Response

Firstly, I would like to congratulate the Authors for presenting such an interesting paper. It is well written and it tries to face the problem of opioid abuse.

However, before the publication, some issues need to be corrected.

1. Naltrexone is an inverse agonist at opioid receptors, and this should be corrected and somehow detailed defined.

Answer: We apologize for that error. This has been corrected. The idea of an inverse agonist and its actions have been more defined too

2. Also, it should be noted that naltrexone may reverse, however not fully, the effects induced by partial agonists.

Answer: This has been added to the text in the mechanism of action section.

3. In the section no. 5 the Authors provided some information regarding pharmacokinetics and/or pharmacodynamics profile of naltrexone, while presenting the examples of naltrexone implants, depending on the country of origin. It would be great if the Authors could demonstrate similarities and differences of the above mentioned in the table, including additional substances used by the producers.

Answer: This is a great idea but the differences aren’t really articulated well in the manuscripts used. One was pallets and the others implants but they were all subcutaneous. The reason why they were different in terms of the formulation was not given in the text so cannot be placed in this manuscript.